# Serum Cytokine Profile, Beta-Hexosaminidase A Enzymatic Activity and GM_2_ Ganglioside Levels in the Plasma of a Tay-Sachs Disease Patient after Cord Blood Cell Transplantation and Curcumin Administration: A Case Report

**DOI:** 10.3390/life11101007

**Published:** 2021-09-24

**Authors:** Alisa A. Shaimardanova, Daria S. Chulpanova, Valeriya V. Solovyeva, Ekaterina E. Garanina, Ilnur I. Salafutdinov, Alexander Vladimirovich Laikov, Vadim V. Kursenko, Lisa Chakrabarti, Ekaterina Yu. Zakharova, Tatiana M. Bukina, Galina V. Baydakova, Albert Anatolyevich Rizvanov

**Affiliations:** 1Institute of Fundamental Medicine and Biology, Kazan Federal University, 420008 Kazan, Russia; AlisAShajmardanova@kpfu.ru (A.A.S.); DaSChulpanova@kpfu.ru (D.S.C.); VaVSoloveva@kpfu.ru (V.V.S.); EEGaranina@kpfu.ru (E.E.G.); IISalafutdinov@kpfu.ru (I.I.S.); AVLajkov@kpfu.ru (A.V.L.); 2Shemyakin-Ovchinnikov Institute of Bioorganic Chemistry, The Russian Academy of Sciences, 19991 Moscow, Russia; 3Individual Entrepreneur Kursenko V.V., 423800 Naberezhnye Chelny, Russia; 567611@rambler.ru; 4School of Veterinary Medicine and Science, University of Nottingham, Nottingham NG7 2RD, UK; lisa.chakrabarti@nottingham.ac.uk; 5Research Centre for Medical Genetics, 115478 Moscow, Russia; doctor.zakharova@gmail.com (E.Y.Z.); tbookina@mail.ru (T.M.B.); gb2003@yandex.ru (G.V.B.)

**Keywords:** lysosomal storage diseases, Tay-Sachs disease, GM_2_ gangliosidosis, β hexosaminidase A, cytokine profile, curcumin, cord blood cell transplantation

## Abstract

Tay-Sachs disease (TSD) is a progressive neurodegenerative disorder that occurs due to a deficiency of a β hexosaminidase A (HexA) enzyme, resulting in the accumulation of GM_2_ gangliosides. In this work, we analyzed the effect of umbilical cord blood cell transplantation (UCBCT) and curcumin administration on the course of the disease in a patient with adult TSD. The patient’s serum cytokine profile was determined using multiplex analysis. The level of GM_2_ gangliosides in plasma was determined using mass spectrometry. The enzymatic activity of HexA in the plasma of the patient was assessed using a fluorescent substrate assay. The HexA α-subunit (HexA) concentration was determined using ELISA. It was shown that both UCBCT and curcumin administration led to a change in the patient’s cytokine profile. The UCBCT resulted in an increase in the concentration of HexA in the patient’s serum and in an improvement in the patient’s neurological status. However, neither UCBCT nor curcumin were able to alter HexA activity and the level of GM_2_ in patient’s plasma. The data obtained indicate that UCBCT and curcumin administration can alter the immunity of a patient with TSD, reduce the level of inflammatory cytokines and thereby improve the patient’s condition.

## 1. Introduction

Tay-Sachs disease (TSD) is a lysosomal storage disease (LSD) which is caused by various mutations in the HexA gene encoding an α-subunit of a β hexosaminidase A (HexA) enzyme. This enzyme specifically cleaves GM_2_ gangliosides (hereinafter GM_2_), which are part of the cell membrane. The HexA enzyme deficiency leads to the accumulation of GM_2_ in various cells and tissues of the body. Nerve tissue is mainly affected due to the high distribution of GM_2_ in neurons. The accumulation of GM_2_ in the cells of the nervous system leads to disability and subsequent death. TSD is characterized by severe neurodegeneration, the onset of an inflammatory response and mental and physical retardation. There are infantile, juvenile and adult forms of TSD, which differ in residual HexA activity, age of onset and severity of the disease [1,2,3,4].

Currently, there are no effective treatments for TSD, and symptomatic therapy is mostly used to support the patients. However, approaches are being developed that can increase the expression of the missing enzyme in order to ease the course of the disease. One of these approaches is the use of curcumin, which has a wide range of therapeutic properties; in particular, it has potential to prevent the development of various neurodegenerative diseases and/or slow their progression [5,6]. To date, there are several studies aimed at studying the molecular mechanisms through which curcumin can affect the level of gene expression of the α- and β-subunits of the HexA enzyme.

For example, the effect of the monocarbonyl analogue of curcumin on the activation of TFEB, which is the main regulator of many cellular processes, such as lysosomal functions, autophagy and membrane repair, has been investigated. It has been shown that curcumin can induce autophagy by directly binding to TFEB and promoting its nuclear translocation. Curcumin-dependent activation of TFEB led to an increase in the expression of the HexA and HexB genes, as well as to an increase in the activity of hexosaminidases and their attraction to the plasma membrane. Based on these data, the monocarbonyl analogue of curcumin may be regarded as a potential drug to treat GM_2_ gangliosidosis, as it leads to an increase in the expression of genes encoding the HexA enzyme and also has the ability to stimulate autophagy, which is generally useful during neurodegeneration [7,8].

Curcumin has been shown to have an inhibitory effect on nonsense-mediated mRNA decay (NMD), which is a process responsible for cleaving mRNAs containing stop codons and/or mRNAs that are not properly spliced. NMD does not allow defective proteins to be synthesized, while curcumin inhibits NMD and influences splicing factors, which in turn leads to an increase in the number of such defective transcripts and contributes to their stabilization. In particular a 1.5-fold increase in the expression of the mutant HexA gene was observed in skin fibroblasts isolated from a TSD patient after cultivation with curcumin (25 μM) for 24 h. It can be suggested that curcumin administration can increase the residual HexA activity in patients by promoting translation of the mutant transcripts [9].

Cell culture studies have shown that due to the ability of curcumin to influence certain molecular pathways, it can increase the expression of the HexA and HexB genes, as well as HexA activity. Additionally, there are numerous studies showing the beneficial effects of curcumin in various neurodegenerative diseases [10,11,12,13,14]. Appendix A provides information on the pharmacokinetics and effectiveness of curcumin in various conditions. However, no investigations have been previously conducted on the effect of curcumin in TSD patients. Thus, it would be interesting to test the hypothesis that curcumin may be useful in the treatment of TSD.

Transplantation of cells expressing a functional enzyme has been shown to prevent further disease progression and help prolong the life of patients [15]. To increase the level of functionally active enzymes in LSD patients, umbilical cord blood cell transplantation (UCBCT) is used. The therapeutic effect is achieved as the umbilical cord blood is a source of stem cells, which constantly express normal enzymes [16,17,18]. In the case of LSDs affecting the nervous system, UCBCT is also relevant, since it is known that blood-isolated immune cells are able to overcome the blood–brain barrier and deliver a normal enzyme to the nervous system [19,20]. For example, in children with Krabbe disease (demyelinating LSD) who received UCBCT before symptoms appeared, normal blood galactocerebrosidase levels, central nervous system myelination and continued developmental skill progress were observed. However, it is important to note that children undergoing transplantation after the appearance of the symptoms had minimal neurological improvements [21,22,23,24,25]. Similar results were obtained after UCBCT in patients with metachromatic leukodystrophy (also demyelinating LSD) [26,27]. UCBCT has also been successfully used for other LSDs [28,29,30,31].

The UCBCT therapeutic effect is probably achieved by cross-correcting the affected neurons. Cord blood cells cross the blood–brain barrier and secrete a functional enzyme that is taken up by nerve cells through endocytosis and enters the lysosomes using mannose-6-phosphate receptors. Thus, cross-correction leads to an increase in the level of the Hex enzyme, clarification of nerve cells from GM_2_ gangliosides and apoptosis arrest [32].

Based on the above findings, we hypothesized that UCBCT and curcumin intake may have a therapeutic effect and improve the condition of the TSD patient. In our study, for the first time, the effect of curcumin and UCBCT on the serum cytokine profile, GM_2_ level and HexA enzymatic activity in the plasma of a patient with adult TSD was investigated.

## 2. Materials and Methods

### 2.1. Isolation of Plasma and Serum from Blood

Whole blood from the TSD patient (*n* = 1) or healthy donors (*n* = 10) was collected in tubes with sodium citrate or a clot activator and then centrifuged at 2000× *g* for 20 min at room temperature. After clotting, the upper phase (plasma or serum) was taken and stored at −80 °C until use. All experiments were carried out in accordance with ethical standards and current legislation (the protocol was approved by the Local Ethics Committee of the Kazan Federal University (No. 24, 22 September 2020). Written informed consent was obtained from each donor. Gender, age and demographics of the patient and healthy donors were the same.

### 2.2. Multiplex Analysis of Cytokine/Chemokine Levels

The MILLIPLEX MAP Human Cytokine/Chemokine Magnetic Bead Panel—Premixed 41 Plex—Immunology Multiplex Assay (#HCYTMAG-60K-PX41 Millipore, Burlington, MA, USA) was used to analyze blood serum samples from the TSD patient and healthy donors (*n* = 10) in accordance with the manufacturer’s recommendations. The kit allows detection of the following analytes: sCD40L, EGF, eotaxin/CCL11, FGF-2, Flt-3 ligand, fractalkine, G-CSF, GM-CSF, GRO, IFN-α2, IFN-γ, IL-1α, IL-1β, IL-1ra, IL-2, IL-3, IL-4, IL-5, IL-6, IL-7, IL-8, IL-9, IL-10, IL-12 (p40), IL-12 (p70), IL-13, IL-15, IL-17A, IP-10, MCP-1, MCP-3, MDC (CCL22), MIP-1α, MIP-1β, PDGF-AA, PDGF-AB/BB, RANTES, TGF-α, TNF-α, TNF-β and VEGF. To determine the concentration of analytes, 25 μL of each sample without dilution was used per well in 3 technical replicates. A standard curve was created for each analyte. Within each standard dilution, the coefficient of variation does not exceed 10%. The data obtained were analyzed using MasterPlex CT software and MasterPlex QT software (MiraiBio Hitachi Software, San Francisco, CA, USA).

### 2.3. Determination of HexA Enzymatic Activity

To analyze the HexA enzymatic activity, a fluorescent substrate 4-methylumbelliferyl-beta-D-N-acetyl-glucosamine-6-sulfate (3.2 mM) (MUGS, TRC, Toronto, ON, Canada) diluted in citrate-phosphate buffer (pH = 4.2) was used. Plasma samples (30 μL) were incubated with 12.5 μL of substrate for 1 hour at 37 °C in opaque 96-well plate. The reaction was stopped by adding 100 μL of glycine carbonate buffer (0.17 M glycine, 0.17 M sodium carbonate) to each well. The fluorescence levels were measured using an Infinite M200Pro reader (Tecan Trading AG, Mannedorf, Switzerland) (Ex = 365 nm, Em = 450 nm). 

### 2.4. Determination of HexA Concentration by ELISA

The concentration of the HexA α-subunit (HexA) in blood serum was determined using ELISA Kit for Hexosaminidase A Alpha (HexA) (SEA195Hu, Cloud-Clone Corp., Houston, TX, USA). According to the protocol recommended by the manufacturer, 100 μL of each sample without dilution in 3 technical replicates was used for analysis. We optimized the method in advance by doing a test run using different plasma dilutions. Samples and standards were added to precoated plates and incubated for 1 h at 37 °C. The liquid was removed, and 100 μL of reagent A was added and incubated for 1 h at 37 °C. The liquid was removed, and the plate was washed 3 times with 1× washing solution. After washing, 100 μL of detection reagent B was added, and plate was incubated for 30 min at 37 °C. The liquid was removed, and the plate was washed 5 times. After that, 90 μL of substrate solution was added, and the plate was incubated for 10–20 min at 37 °C (no more than 30 min) in the dark. Next, the reaction was stopped by adding 50 μL of stop solution. The optical density was measured using an Infinite M200Pro reader (Tecan Trading AG, Mannedorf, Switzerland) at 450 nm.

### 2.5. Mass Spectrometry Analysis of GM_2_ Plasma Levels

The method of determination of gangliosides in the patient’s plasma was based on [33,34,35]. The GM_2_ 18:1/18:0 standard (Cat. No.:1502) and its isotopically labeled analogue GM_2_-*d3* 18:1/18:0 (Cat. No.:2051) were used to refine the methodology and construct calibration curves (Matreya LLC, State College, PA, USA).

Plasma extracts were obtained by precipitation of proteins with methanol. The GM_2_-*d3* standard was added to 50 μL of plasma with a final concentration of 1 μg/mL. Subsequently, 200 μL of methanol was added and the mixture was stirred for 1.5 min at 2500 rpm. The mixture was centrifuged for 15 min at 16,000× *g*, and the supernatant was transferred into a chromatographic vial for analysis on a QTRAP 6500 mass spectrometer (Sciex, Singapore) combined with high-performance liquid chromatography using Infiniti 1290 chromatograph (Agilent, Waldbronn, Germany). The samples were separated on Discovery^®^ HS C18 column, 5 cm × 2.1 mm, sorbent 3 μm. Chromatographic phase A was composed of 95% water, 5% methanol, 5 mM ammonium acetate; phase B—50% methanol, 50% isopropanol, 5 mM ammonium acetate. Gradient: 0–1 min, 30% phase B; 1–2 min, gradient up to 80% phase B; 2–7 min, gradient up to 95% phase B; 7–9 min, 95% phase B; 9–9.1 min, gradient up to 30% of phase B; 9.1–12 min, 30% of phase B. Flow rate was 0.4 mL/min, and column temperature was 40 °C. Mass spectrometer source parameters: curtain gas: 35 psi; IonSpray voltage: 5200 V; temperature: 500 °C; gas 1: 60 psi; gas 2: 60 psi. Detection of negatively charged ions was carried out according to Table 1. The calibration curve and the GM_2_ concentration in the samples were determined from the area ratio of light to heavy peptides.

### 2.6. Statistical Analysis

Statistical analysis was achieved using GraphPad Prism 7 software (GraphPad Software, San Diego, CA, USA), with one-way ANOVA followed by Tukey’s HSD post hoc comparison test. Each patient sample (mean of technical replicates) was compared with both the mean of 10 healthy donors and with other patient samples. Significant probability values are denoted as * *p* < 0.05, ** *p* < 0.01 and *** *p* < 0.001, **** *p* < 0.0001.

## 3. Case Presentation

### 3.1. Patient Information

Patient X, born in 1990 (30 years old), was diagnosed with the adult chronic form of TSD (OMIM 272800), which was confirmed by the results of enzyme activity and DNA sequence diagnostics. Compound heterozygous mutations in the HexA gene were identified in the patient. The first mutation c805G > A (p.Gly269Ser) has already been described in the literature and is one of the common variants causing the adult form of TSD among Ashkenazi Jews. The second mutation, 346 + 2dupT, is new and has never been described as pathogenic before. It was firstly identified in our patient. It was found that the second mutation causes the disruption of RNA splicing, resulting in the absence of a second exon that encodes part of the beta-hexosaminidase domain required for HexA activity.

Perinatal pathology of the central nervous system (CNS) was diagnosed in childhood, and disability gradually increased until the age of 20 when the patient lost the ability to move independently. There is an acute tetraparesis up to plegia in the left leg, gross dynamic and static abnormalities, speech impairment and impaired movement and self-care ability; the patient needs ongoing supportive care. 

The data of magnetic resonance imaging did not show signs of the expansive process (t-r) and focal pathology of the brain. However, atrophy of all parts of the cerebellum and hypoplasia of the lower worm were revealed. As a result of electroneuromyography, degenerative processes in motor neurons were revealed.

The patient underwent transplantation of single-group and Rh-compatible cord blood cells without side effects (total number of leukocytes was 3 × 10^8^ (3.3 × 10^6^/kg), the cells were obtained at the Samara Cord Blood Bank with WMDA accreditation, Samara, Russia). To prevent graft-versus-host disease (GVHD), methylprednisolone and cyclosporine were prescribed for 2 months.

Based on existing studies showing the therapeutic effects of curcumin in neurodegenerative diseases, as well as its ability to increase the expression of HexA enzyme genes, a course of curcumin (Curcumin C3 Complex, California Gold Nutrition, Los Angeles, CA, USA) was prescribed for the patient. The patient was given curcumin for 3 months according to the following regimen: 500 mg/day for 10 days, 1000 mg/day for 10 days and then 3500 mg/day for 100 days.

Before the UCBCT, the patient had normal consciousness. However, the following syndromes were observed in the patient: chronic fatigue, acute central tetraparesis to plegia in the left leg, gross dynamic and static abnormalities without the ability to self-care and move without help, speech disorders and swallowing difficulties.

Within one month after the UCBCT, the following changes in neurological status were observed: a decrease in the degree of paresis in the lower limbs, a decrease in the severity of athetosis and limb tremor, a significant improvement in speech and the patient was able to move with single-sided support and independently transfer from bed to a wheelchair. The general state of the patient and daily activity were improved, as it was noted by the patient’s family members. In addition, the quantitative analysis of limb paresis showed an increase in muscle strength after UCBCT (Table 2).

While taking curcumin, the patient’s state was stable. No changes in neurological status with a tendency to decrease in neurological deficit were observed and persisted for 1 year; after this time, the neurological deficits gradually began to return.

### 3.2. Dynamic Changes in HexA Activity and the GM_2_ Level in the TSD Patient’s Plasma

It has been shown that HexA activity was not significantly altered by UCBCT or curcumin administration (Figure 1a). On average, the enzymatic activity in the patient’s plasma was 4.6 ± 0.6 nmol/mL/h, which is 5.3 times lower than the lower limit of the normal rage. However, a statistically significant increase in patient’s serum HexA concentration was found after UCBCT (Figure 1b). The HexA concentration after UCBCT was increased up to 8.52 ng/mL, which is 5 times higher compared to the average HexA concentration before UCBCT (1.74 ± 0.33 ng/mL). It has also been shown that the concentration of the enzyme in the patient’s serum was within the range of the control group of healthy donors. The level of GM_2_ in the patient’s plasma and in the plasma of healthy donors (*n* = 10) was determined using mass spectrometry. The average value of the patient’s GM_2_ level (255.9 ± 52.6 ng/mL) was not statistically different from the GM_2_ level observed in healthy donors (315.6 ± 107.9 ng/mL). UCBCT or curcumin administration also failed to affect GM_2_ levels (Figure 1c).

### 3.3. Cytokine Profile

The levels of cytokines in the serum of healthy donors and the patient over time were determined. The level of each analyte was compared before and after transplantation, before and after long-term use of curcumin and with the results of the control group. It was shown that before UCBCT and curcumin administration, the levels of VEGF, EGF, eotaxin-1, MDC, sCD40L and IL-8 in the patient’s serum were significantly higher compared to the control group of healthy donors (*n* = 10). VEGF level was 2.8 times higher (387.5 ± 29.8 pg/mL), EGF (345.4 ± 0.8 pg/mL) was 3.09 times higher, eotaxin-1 (200.6 ± 1.9 pg/mL) was 3.1 times higher, MDC (1127.9 ± 5.7 pg/mL) was 2.7 times higher, sCD40L (10260.3 ± 793.1 pg/mL) was 2.2 times higher and IL-8 (21.3 ± 1.3 pg/mL) was 2.7 times higher in comparison with healthy donors (VEGF (138.2 ± 133.7 pg/mL), EGF (111.6 ± 103.4 pg/mL), eotaxin-1 (64.8 ± 28.7 pg/mL), MDC (413.3 ± 175.8 pg/mL), sCD40L (4576.1 ± 3168.7 pg/mL), IL-8 (7.8 ± 6.0 pg/mL)) (Figure 2, Appendix A).

Analysis of changes in the cytokine levels over time showed that the levels of PDGF-AA, eotaxin-1, sCD40L and IP-10 were significantly reduced after curcumin administration. By the end of the prescribed course of curcumin, the levels of PDGF-AA (577 ± 30 pg/mL), eotaxin-1 (93.1 ± 17.5 pg/mL), sCD40L (8345.5 ± 393.2 pg/mL) and IP-10 (28.8 ± 2.3 pg/mL) were decreased by 4.7, 2.1, 1.2 and 2.8 times, respectively, compared with the starting point of the curcumin administration (PDGF-AA (2715 ± 74.5 pg/mL), eotaxin-1 (200.6 ± 1.9 pg/mL), sCD40L (10,260.3 ± 793.1 pg/mL) and IP-10 (80.4 ± 2.7 pg/mL)) (Figure 2).

EGF and IL-8 levels were also significantly reduced after UCBCT. The level of EGF was 2.3 times decreased after UCBCT (153.4 ± 16.9 pg/mL), compared to the analysis before UCBCT (345.4 ± 0.8 pg/mL), and the IL-8 level was 2.6 times decreased after UCBCT (8.1 ± 0.2 pg/mL), compared to the point before transplantation (21.3 ± 1.3 pg/mL) (Figure 2).

## 4. Discussion

### 4.1. UCBCT and Curcumin Administration Fail to Affect GM_2_ Levels in the Patient’s Plasma

Gangliosides consist of a ceramide and an oligosaccharide connected to the residues of N-acetylneuraminic acid. There are gangliosides containing one (GM), two (GD), three (GT) and four (GQ) sialic acid residues. Monosialogangliosides (GM) are classified based on the amount of sugar residues in the oligosaccharide area of the molecule: GM_1_—four sugar residues, GM_2_—three and GM_3_—two [36]. In addition, within each class, the gangliosides may differ in the length of the chain and the amount of fatty acid double bonds, which make up the ceramide [35].

Variations in the molecular weight of ceramide (18:1/14:0, 18:1/16:1, 18:1/16:0, 18:1/18:1, 18:1/18:0, 18:1/20:0), which could be detected in plasma GM_2_, can be used as metabolic markers of TSD [35]. However, our results showed that the assessment of the ganglioside level in the patient’s blood plasma does not correlate well with the metabolic status of TSD patients. Analysis of the dynamic changes of GM_2_ level did not show a statistically significant difference between the patient and the group of healthy donors. Perhaps this is due to the fact that gangliosides are mostly accumulated inside the cells but not in the bloodstream. Based on this, the metabolic status of patients with GM_2_ gangliosidoses should be analyzed in other biological materials: for example, in mononuclear cells, fibroblasts [34] or CSF [33].

### 4.2. UCBCT and Curcumin Administration Fail to Affect the Activity of HexA in the Patient’s Plasma, but Patient’s Serum HexA Concentration Increases after UCBCT

We investigated the influence of UCBCT and the curcumin administration on the plasma HexA enzymatic activity and serum HexA concentration. We found that the activity of HexA in the plasma remains unchanged both after UCBCT and the course of curcumin administration. However, a significant increase in patient’s serum HexA concentration after UCBCT was observed. It has also been noted that the patient’s serum HexA concentration was within the range of the control group. This is because the mutation results in altered HexA activity but not a decrease in its amount. Curcumin intake does not change either the activity or the concentration of the enzyme despite the fact that earlier studies have shown that curcumin is able to increase the level of expression and the activity of the HexA enzyme in vitro [7,9]. One explanation could be that the allowable oral administration dose is insufficient to increase the level of the enzyme in plasma.

### 4.3. The Levels of VEGF, EGF, Eotaxin-1, MDC, sCD40L and IL-8 Are Elevated in the Blood Serum of the Patient with TSD

In this study, we analyzed, for the first time, the cytokine profile in the serum of a TSD patient and observed a significant increase in the levels of VEGF, EGF, eotaxin-1, MDC, sCD40L and IL-8 in the patient’s serum. As mentioned earlier, previous studies were aimed at studying the cytokine profile in the CSF of patients with infantile and juvenile forms [37,38], whilst we analyzed cytokines in the serum of a patient with adult TSD. Though CSF analysis would be more informative since the TSD diseases predominantly affect the CNS, we have examined the patient’s serum, since we wished to investigate dynamic changes. The serum collection for the diagnosis and screening of the effectiveness of treatment is more readily performed, which is undoubtedly an advantage in the clinic.

We divided the cytokines into three groups depending on their functions: leukocyte chemoattractants (IL-8), regulators of angiogenesis (VEGF and EGF) and immunoregulatory cytokines (eotaxin-1, MDC and sCD40L) [39].

An increase in the level of angiogenic factors VEGF and EGF in the serum of the patient was observed. Changes in the level of VEGF secretion were found in some LSDs that do not affect the CNS [40,41]. Unfortunately, the function of EGF in the pathogenesis of LSDs is still unclear. It is known that EGF mediates an effective reparative response to any attack to biophysical integrity and also participates in the regulation of pro-inflammatory activation [42]. We suggest that VEGF and EGF levels could be increased in response to the cellular and tissue damage caused by the accumulation of GM_2_.

The secretion of immunoregulatory cytokines, such as eotaxin-1, MDC and sCD40L, was also elevated in the serum of the TSD patient. It is assumed that increased eotaxin levels are associated with degenerative processes in the CNS [43,44]. MDC is a chronic mediator of inflammation [45]. sCD40L plays an important role in T-cell-dependent activation of B cells [46] and chronic inflammation [47,48]. Observed increased levels of immunoregulatory cytokines (eotaxin-1, MDC and sCD40L) in the TSD patient most likely indicate pathological processes associated with neurodegeneration and neuroinflammation.

The level of IL-8, which is a chemotactic factor, was also enhanced. The IL-8 level was elevated in the CSF of patients with metachromatic leukodystrophy [49] and mucopolysaccharidosis type I-H [50] and in the serum of patients with Gaucher disease [51,52]. 

Thus, an increase in the level of VEGF and EGF growth factors may be the physiological response to damage of different tissues due to the GM_2_ accumulation, and the levels of immunoregulatory cytokines eotaxin-1, MDC, sCD40L and chemoattractant IL-8 may be increased due to neurodegeneration and inflammation of the CNS.

### 4.4. EGF and IL-8 Levels Decrease in the TSD Patient’s Serum after UCBCT

Cord blood is rich in hematopoietic stem cells and progenitor cells. UCBCT is widely used for patients who do not have a compatible HLA donor for bone marrow transplantation (BMT) [53]. BMT and UCBCT donor cells produce and thereby restore levels of reduced or missing enzymes, thus providing therapy for LSDs [15,54]. 

Human umbilical cord blood is an important source of stem cells and progenitor cells capable of exerting a neuroprotective effect in degenerative processes caused by various factors. The therapeutic effect of UCBCT in TSD is mediated by the ability of the umbilical cord blood cells to express HexA and migrate into the nervous system. The normal enzyme expressed by donor cells can be delivered to the patient’s nerve cells through endocytosis [32].

In this paper, the cytokine profile of a patient with an adult form of TSD after UCBCT was evaluated. We compared the cytokine profile before and 18 weeks after transplantation. After UCBCT, the EGF and IL-8 levels in the patient’s serum were significantly reduced. As mentioned earlier, the levels of these analytes were significantly increased in this patient before transplantation compared to the control group and therefore likely play a role in the pathogenesis of the disease. It can be assumed that UCBCT influenced the level of these analytes.

Investigation of cytokine profiles in adult patients after a single UCBCT has shown that about 80% of patients suffer from pre-engraftment immune reaction after UCBCT, which can be due to a cytokine storm. The levels of certain cytokines, especially IL-5 and IL-6, were strongly increased in the first 8 weeks after transplantation, and after 8 weeks, cytokine levels and the immune response were mostly normalized. Therefore, it is important to analyze the effect of UCBCT more than 2 months after transplantation [55].

Our data confirm that the IL-8 level decreases after UCBCT. For example, it was shown that 1 month after UCBCT, the IL-8 level in the blood serum initially increased and then decreased. In samples taken 4.5 months after transplantation, the cytokine profile showed a decrease in IL-8 levels. A negative correlation of IL-8 level with the amount of CD14^+^ and CD20^+^ cells has been described previously [56]. Additionally, administration of umbilical cord-isolated MSCs in animals with acute radiation disease has been shown to decrease serum IL-8 levels [57].

Changes in the level of EGF after UCBCT have not been previously investigated. However, since EGF is secreted in response to impaired tissue integrity, it could be that the enzyme produced by donor cells reduces the GM_2_ accumulation, which can reduce damage in the nervous system, which in turn leads to a decrease in the EGF level [58,59].

### 4.5. PDGF-AA, Eotaxin-1, IP-10 and sCD40L Levels Are Reduced in the TSD Patient’s Serum after Administration of Curcumin

Many studies have aimed to evaluate the therapeutic potential of curcumin. It has been shown that curcumin is able to suppress inflammatory responses by influencing the level of inflammatory cytokines and can affect the progression of many neurodegenerative diseases [60]. In addition, curcumin has been shown to increase the expression of HexA and HexB genes in vitro [7,9].

In this study, we investigated the influence of curcumin on the dynamic changes of the cytokine profile in the TSD patient. The patient took curcumin for 3 months as follows: 500 mg/day for 10 days, 1000 mg/day for 10 days and then 3500 mg/day for 100 days.

Since there are no investigations aimed at studying the influence of curcumin on TSD patients, we compared our results with studies of other related LSDs and neurodegenerative diseases. Some other LSDs, such as TSD, have neurological manifestations, including cognitive impairments. Each analyte that had a significant difference before and after curcumin administration will be discussed separately.

The level of platelet growth factor PDGF-AA was decreased in the patient’s serum by the end of the curcumin course. PDGF-AA has angiogenic properties and is released during vessel damage; it is also an oligodendrocyte precursor mitogen. It has been shown that curcumin can affect the PDGF signal pathways, but has no direct effect on PDGF expression. For example, curcumin inhibits PDGF-stimulated migration, proliferation and synthesis of vascular smooth muscle cell in vitro [61]. Curcumin also inhibits PDGF-induced proliferation of human hepatic myofibroblasts [62]. It can also have a connection with nervous system diseases: for example, the development of severe demyelization due to PDGF-AA deficiency has been described [63].

We also observed a decrease in the immunoregulatory cytokines eotaxin-1 and sCD40L as a result of curcumin administration. Before taking curcumin, eotaxin-1 levels were significantly higher compared to controls; after the course of curcumin administration, the level decreased when compared with the control group. As mentioned earlier, elevated levels of eotaxin correlate with a number of nervous system disorders. To summarize, eotaxin is elevated in the cerebellum of model mice of Niemann–Pick type C (NPC) [64], is negatively correlated with gray matter in patients older than 60 years with late-life major depression [65] and is possibly associated with the loss of dopaminergic neurons in patients with Parkinson’s disease [66]. In a mouse model of allergic asthma, it was shown that treatment with curcumin leads to suppression of the expression of many inflammatory cytokines, including eotaxin [67]. Similar results were obtained in a mouse model of chronic asthma, where intranasal administration of curcumin significantly reduced eotaxin levels [68].

The level of sCD40L in the patient’s serum was significantly higher compared to the control group. After curcumin administration, the sCD40L level was decreased to the normal values of the control group. As described above, CD40L is involved in the pathogenesis of inflammatory diseases [46,47]. Previously, it has been shown that after 8 months of curcumin administration, sCD40L levels were significantly reduced in 100 patients with osteoarthritis [69]. This coincides with our observation of a decrease in the sCD40L level in the serum after curcumin administration.

After curcumin administration, the IP-10 chemoattractant level was significantly reduced. IP-10 is a chemokine involved in the attraction of immune cells at inflammation sites [70], and IP-10 levels are frequently elevated during CNS infection [71]. High levels of IP-10 are associated with some LSDs [64,72,73]. Our data and other studies suggest that curcumin suppresses the expression of IP-10 [74,75], which could mean that it may help to reduce inflammation.

## 5. Conclusions

In the present work, the influence of UCBCT and curcumin administration on the cytokine profile, HexA activity and GM_2_ level in a patient with an adult form of TSD was investigated. The patient’s serum levels of VEGF, EGF, eotaxin-1, MDC, sCD40L and IL-8 were significantly higher compared to the control group. A decrease in the levels of IL-8 and EGF after UCBCT and decreases in EGF, PDGF-AA, PDGF-AA/AB, eotaxin-1, sCD40L and IP-10 in the serum after curcumin administration were revealed. The data show that the UCBCT and curcumin administration affect the patient’s cytokine profile. Additionally, UCBCT leads to an increase in the concentration of HexA in the serum and an improvement in the patient’s neurological status. We suggest that as a result of UCBCT, the functional enzyme produced by donor cells can reduce the number of accumulated GM_2_, which can reduce tissue damage. In addition, the immunoregulatory properties of cord blood cells may ensure the reduction of inflammation. Curcumin has an anti-inflammatory and antioxidant effect, which may lead to a decrease in the level of inflammatory cytokines. However, neither UCBCT nor the curcumin administration affected the activity of HexA and the level of GM_2_ in the patient’s plasma. Thus, in this case report, we have investigated UCBCT and curcumin supplementation as potentially useful approaches for the relief of a TSD patient’s condition. For more accurate conclusions, studies with a large number of participants are required.

## Figures and Tables

**Figure 1 life-11-01007-f001:**
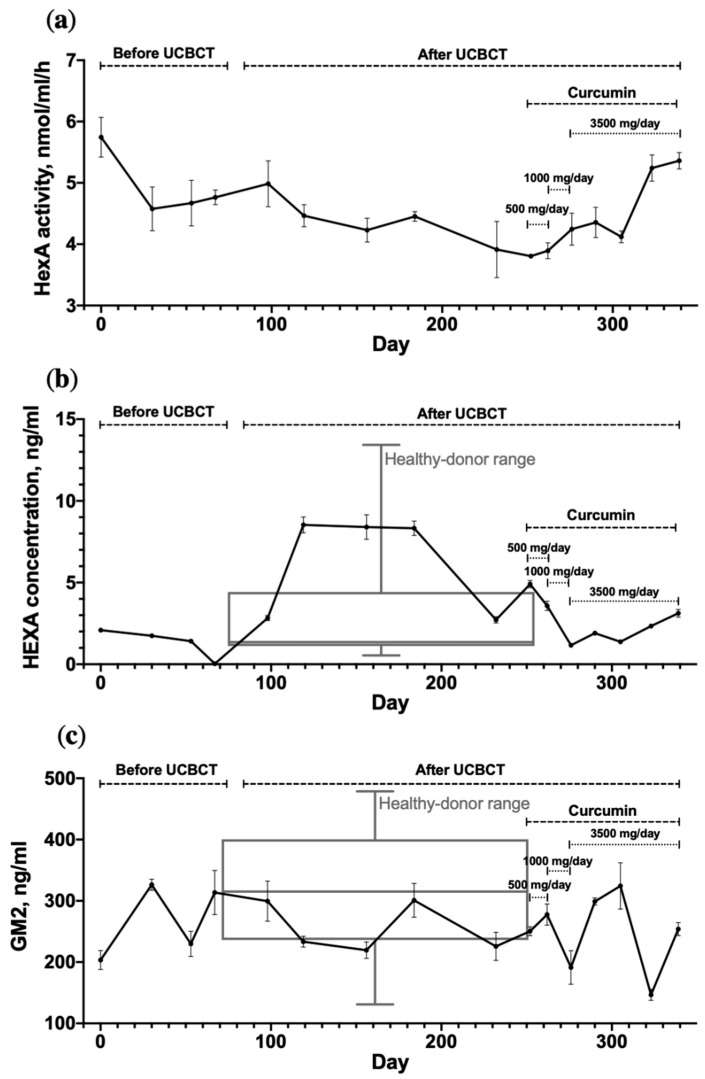
(**a**) Dynamic changes in the HexA activity in the TSD patient’s plasma after UCBCT and curcumin administration; normal level of HexA activity observed in healthy donors is 24–148 nmol/mL/h; (**b**) Dynamic changes in the HexA concentration in the TSD patient’s serum after UCBCT and curcumin administration. The HexA concentration range measured in a cohort of healthy donors (*n* = 10) is shown in gray; (**c**) Dynamic changes in the GM_2_ level in the TSD patient’s plasma after UCBCT and curcumin administration. The GM_2_ level range measured in a cohort of healthy donors (*n* = 10) is shown in gray. The patient took curcumin for 3 months as follows: 500 mg/day for 10 days, 1000 mg/day for 10 days and then 3500 mg/day for 100 days. The graph represents the mean and standard deviation of three technical replicates of each patient point and the mean and standard deviation of one technical replicate of ten healthy donors (ten biological replicates). Statistical analysis was achieved using GraphPad Prism 7 software (GraphPad Software), with one-way ANOVA followed by Tukey’s HSD post hoc comparison test.

**Figure 2 life-11-01007-f002:**
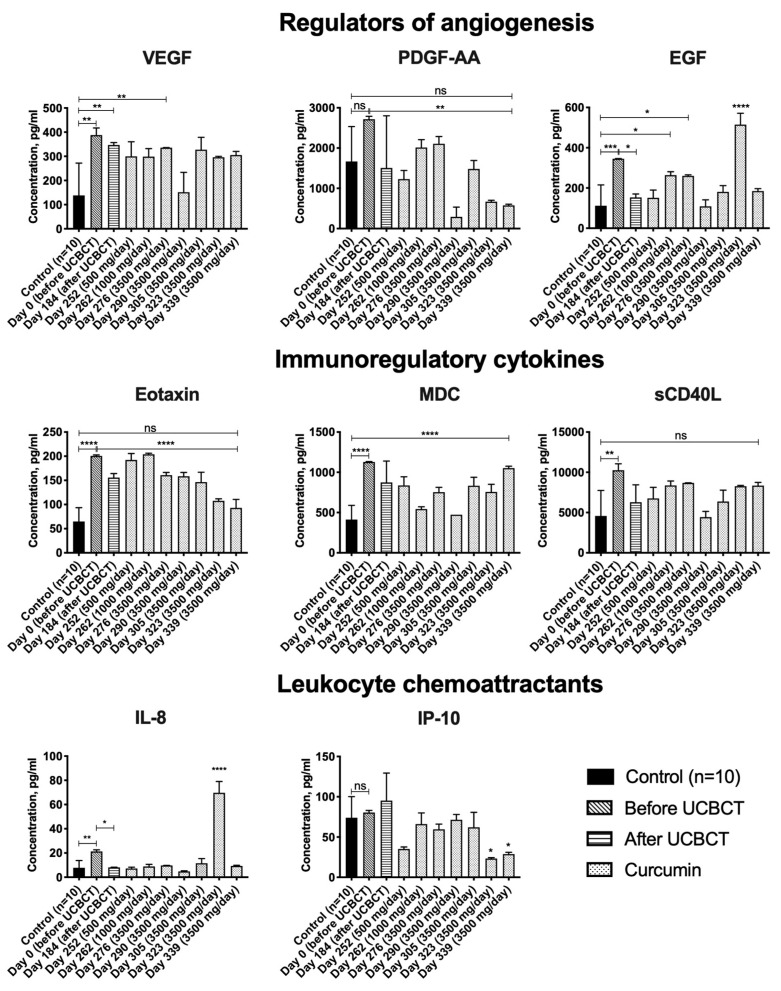
Changes in the cytokine levels in the TSD patient’s serum after UCBCT and after the curcumin administration. The patient took curcumin for 3 months as follows: 500 mg/day for 10 days, 1000 mg/day for 10 days and then 3500 mg/day for 100 days. The graph represents the mean and standard deviation of three technical replicates of each patient point and the mean and standard deviation of one technical replicate of 10 healthy donors (10 biological replicates). Statistical analysis was achieved using GraphPad Prism 7 software (GraphPad Software), with one-way ANOVA followed by Tukey’s HSD post hoc comparison test. Significant probability values are denoted as * *p* < 0.05, ** *p* < 0.01 and *** *p* < 0.001, **** *p* < 0.0001.

**Table 1 life-11-01007-t001:** MRM parameters.

Q1	Q3	Fragment	Declustering Potential	Entrance Potential	Collision Energy
1382.785	290.1	* GM2_1	−175	−10	−70
1382.785	1091.5	GM2_2	−175	−10	−72
1382.785	888.6	GM2_3	−175	−10	−70
1382.785	564.3	GM2_4	−175	−10	−92
1385.723	290.0	* GM2_d3_1	−198	−10	−68
1385.723	1094.5	GM2_d3_2	−198	−10	−64
1385.723	567.6	GM2_d3_3	−198	−10	−84
1385.723	891.5	GM2_d3_4	−198	−10	−76
1385.723	729.5	GM2_d3_5	−198	−10	−82

* Fragment used for quantitative analysis of GM_2_.

**Table 2 life-11-01007-t002:** Quantitative analysis of limb paresis.

Q1	before the UCBCT	6 Months after the UCBCT
Strength of the arm muscles	4 points	4 points (significant reduction in tremor and dynamic ataxia was noted)
Strength of the proximal muscles of the left leg	1 point	2.5 points
Strength of the proximal muscles of the right leg	2 points	2.5 points
Strength of the distal muscles of the left leg	2.5 points	3 points
Strength of the distal muscles of the right leg	3 points	3 points

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
