# Peer review of "Serum Cytokine Profile, Beta-Hexosaminidase A Enzymatic Activity and GM2 Ganglioside Levels in the Plasma of a Tay-Sachs Disease Patient after Cord Blood Cell Transplantation and Curcumin Administration: A Case Report"

_life, 2021, doi:10.3390/life11101007_

Round 1
Reviewer 1 Report
The work by Shaimardanova and colleagues describes a case report of a single patient affected by Tay-Sachs disease (TSD) and treated with umbilical cord blood cell transplantation (UCBCT) and curcumin administration. Authors report the effects of treatments on the enzymatic activity and concentration of HexA, GM2 gangliosides levels, patient's serum cytokine profile, and neurological status.
Certainly, the combined use of UCBCT and curcumin in a TSD patient is original, although I have some concerns regarding the results obtained.
Major points:
- The measured effects are attributed to UCBCT or curcumin administration based on the response over time, but an evaluation of the cross effects between treatments is missing. Which are the systemic peculiar effects of the massive administration of curcumin? Are known other TSD cases that have undergone UCBCT and what are the effects? I think that ad hoc studies should be conducted on preclinical models before testing treatments on patients;
- Authors report an increase in the concentration of Hexa after UCBCT but not in its enzymatic activity. Can authors further comment on this data? Why after transplantation there is an increase of protein concentration but not in its activity? What is the genetic profile of the patient and which are the effects of the mutations in the encoded protein?
- About cytokines profiles (Fig. 1), a timeline describing the days of UCBCT and curcumin administration would be useful (as made in Fig. 2). However, again, I'm not convinced that some cytokines changes are due only to curcumin administration or if these are the results of a cross-effect.
Minor points:
- Introduction: The first part of the introduction relating to the description of TSD would benefit from additional bibliographic references as the two reported belong to authors themself.
- Introduction: I find the rationale for using curcumin on a TSD patient not very thorough. Although there is some evidence about the benefits of curcumin on neurological patients, the preliminary data described focuses only on two in vitro studies and would instead deserve more attention.
- Discussion: The discussion on the serum cytokine profile is very long and particularly elaborate. I suggest first describing the effects on Hexa and gangliosides and then inserting the paragraphs on cytokines (and doing the same for result sections).
- Conclusions: The conclusions "Overall, our data support the recommendation of UCBCT and curcumin in patients with TSD" seem particularly pretentious, especially since the study is conducted on a single patient. I suggest reconsidering them.
Reviewer 2 Report
In this paper, Alisa A. Shaimardanova investigated the potential therapeutic effect of the TSD patient after UCBCT and curcumin intake, which was supported by a range of angiogenesis and cytokine factors as measured by MS. The results are sound and could contribute potential applications in clinics.
1. As major variants were observed in the control groups of each panels of fig1, does the authors checked the "outliner(s)" of the data set, if yes, these data can be removed and give better statistics.
2. It's hard to get a clear conclusion from the fig1 as presented in this output, since you have increase or decrease that was not consistent with each other, any better interpretation of the data?
Reviewer 3 Report
The case report titled ‘Serum cytokine profile, beta-hexosaminidase A enzymatic activity and GM2 ganglioside levels in the plasma of a Tay-Sachs disease patient after cord blood cell transplantation and curcumin administration: a case report’ presents an interesting finding that UCBCT helps in the treatment of Tay-Sachs disease. However, I have the following comments that should be addressed.
- Abstract: Sometimes it is written as ‘HEXA’ and sometimes as ‘hexA’. I believe that the authors mean enzymatic activity or protein levels in both the cases. Please be consistent in naming the protein.
Line 22: ‘HEXA concentration was determined’- there is a typo. Determined is written as determent.
- Line 101: Do you mean based on the above findings…?
- Materials and methods:
Line 108: how many TSD patients and healthy donors were included in the study?
Section2.2: What volumes of samples were used per well? In replicates? Were the samples diluted? If the samples were undiluted, there may be artefacts in the measurements. Sometimes plasma or serum gives high values of the measured protein falsely. Was the kit used for measuring the cytokines partially validated for dilutional linearity, spike recovery, intra and inter-assay %CVs to test the matrix effects of the samples and the reproducibility of the kit?
Section 2.4: The same questions raised in section 2.2 also arise here with HexA ELISA kit. Please comment.
- Case presentation:
Section 3.2: What were the clinical characteristics/demographics of the healthy donors? Were they age matched with the TSD patient? Were all healthy donors ‘age-matched males’ similar to the patient?
Figure 1: How was the ANOVA used is a bit confusing. Each day after treatment of the patient was compared with the mean of 10 healthy donors measured on a single day? 9 days of patient compared with one day of healthy donors (mean)? 9 days are also compared with each other?
Please mention clearly in Figure1, which plots are for Curcumin?
Lines 237-238: As can be seen from Figure 1, the levels of CD40L are actually increased and not reduced after Curcumin administration. Please mention that you are e.g. comparing day 339 of administration with day 252 of Curcumin administration and so on to make it clear what is compared with what.
- Did HexA concentration increase after UCBCT have any effect on the disease symptoms (as shown in Table 2)? This should be mentioned along with the result on HexA concentration change. These results might be combined together in a single plot.
- In the discussion section, the repeated use of the phrase ‘The authors’ is confusing. It is not clear whether the associated results mentioned in the discussion are from previous studies or results from the current study.
- Comparing just one TSD patient with controls is not sufficient to call the altered proteins as ‘biomarkers’ for the disease. The authors should be careful in using this term. Moreover, the authors should re-structure the discussion. Please start with the most significant results from the current study and then draw parallels with previous studies. For example, lines 317-325 should be the starting paragraph for the discussion section.
- All sub-sections within the discussion section are too long, especially 4.1. These can be compressed at least to half and only the most relevant literature should be mentioned here.
- The possible biological mechanism how HexA level improvement after UCBCT may improve disease symptoms should be emphasized in the discussion section.

Round 2
Reviewer 1 Report
The authors addressed my concerns accordingly and the manuscript can be accepted for publication.
Author Response
Thank you very much for your previous comments that helped us improve this manuscript. We have also included the following changes in the main manuscript text:
1) To determine the concentration of analytes, 25 μl of each sample without dilution were used per well in 3 technical replicates. A standard curve was created for each analyte. Within each standard dilution, the coefficient of variation does not exceed 10% (Lines 126-129).
2) According to the protocol recommended by the manufacturer, 100 μL of each sample without dilution in 3 technical replicates were used for analysis. We optimized the method in advance by doing a test run using different plasma dilutions (Lines 143-145).
3) Each patient sample (mean of technical replicates) was compared with both the mean of 10 healthy donors and with other patient samples (Lines 181-183).
Reviewer 3 Report
Please include the changes mentioned in your responses in the main manuscript text.
Author Response
Thank you very much for your previous comments that helped us improve this manuscript. We have included the following changes in the main manuscript text:
1) To determine the concentration of analytes, 25 μl of each sample without dilution were used per well in 3 technical replicates. A standard curve was created for each analyte. Within each standard dilution, the coefficient of variation does not exceed 10% (Lines 126-129).
2) According to the protocol recommended by the manufacturer, 100 μL of each sample without dilution in 3 technical replicates were used for analysis. We optimized the method in advance by doing a test run using different plasma dilutions (Lines 143-145).
3) Each patient sample (mean of technical replicates) was compared with both the mean of 10 healthy donors and with other patient samples (Lines 181-183).